# OpenReview forum: "The Troubling Emergence of Hallucination in Large Language Models - An Extensive Definition, Quantification, and Prescriptive Remediations"
_EMNLP/2023/Conference — EMNLP 2023 Main_

### Official Review · Reviewer_L4uh · 2023-08-01

**Soundness:** 3

**Excitement:**

4: Strong: This paper deepens the understanding of some phenomenon or lowers the barriers to an existing research direction.

**Paper Topic And Main Contributions:**

The paper addresses the phenomenon of hallucinations in LLMs. The authors begin with categorizing hallucinations according to three aspects: degree, orientation and category, with elaborate explanations and examples for each aspect. The authors then crowdsource the first publicly available hallucination dataset, comprising of 75,000 text prose entries, generated by 15 LLMs, where hallucinations are marked and accompanied by degree, orientation and category labels. The authors also introduce a novel evaluation for hallucination (HVI). Lastly, the authors propose two strategies for hallucination mitigation – an automatic approach, involving high-entropy words identification, and a human-in-the-loop approach.
The main contributions of this paper are its hallucination categorization scheme, its novel labeled hallucination dataset and the devised hallucination metric.

**Reasons To Accept:**

1.	The proposed hallucination categorization scheme, as well as the released dataset, are important steps towards a better characterization and handling of this phenomenon.
2.	Their proposed novel metric is also an important step in the research of hallucinations in LLMs.

**Reasons To Reject:**

1.	The paper lacks an investigation of the utility of the HVI metric, compared to other metrics, as well as its correlation with human judgement.

**Reproducibility:**

4: Could mostly reproduce the results, but there may be some variation because of sample variance or minor variations in their interpretation of the protocol or method.

**Reviewer Confidence:**

4: Quite sure. I tried to check the important points carefully. It's unlikely, though conceivable, that I missed something that should affect my ratings.

---

> ### Author Rebuttal · Authors · 2023-08-28
>
> Kindly consult the general feedback regarding the "long abstract and long appendix," as well as the critique on mitigation techniques in the rebuttal for Reviewer #1. We will also add the necessary parts in the main paper with the extra granted page post-acceptance.
>
> HVI metric, compared to other metrics, as well as its correlation with human judgment.
>
> --------------------------------------------------------------------------------------------------------------------------------------------------------
>
> HVI computation relies on manually annotated data, leaving us uncertain about how to effectively contrast it with human assessment. Any specific recommendations in this regard would be greatly appreciated. Currently, we are not acquainted with alternative metrics/indices for quantifying hallucination. If there are any references to such metric/index available beyond our current knowledge, we would greatly appreciate it if you could provide them. This would allow us to examine and compare them with HVI.

---

### Official Review · Reviewer_rKxV · 2023-08-04

**Typos Grammar Style And Presentation Improvements:** 1. The abstract it too long. You don’…
**Soundness:** 3

**Excitement:**

2: Mediocre: This paper makes marginal contributions (vs non-contemporaneous work), so I would rather not see it in the conference.

**Missing References:**

I don't think there are missing references, but the news articles, public statements and reference to Wikipedia pages that are mentioned throughout the paper should not be treated as standard academic citations. You should simply include the URL in a footnote.


**Paper Topic And Main Contributions:**

In this paper the authors introduce a new dataset of 75K texts that were generated by 15 different LLMs, manually annotated for different types of hallucinations which they defined using a new taxonomy as part of this paper. The texts are broken into sentences, and every sentence is assigned with only one label (although potentially it may contain several types of hallucinations).

**Reasons To Accept:**

The new dataset is valuable and I am sure it will be useful for the EMNLP research community. I like the long and informative introduction section.

**Reasons To Reject:**

While the paper is well written, and the motivation behind this project is provided with many details, I do see some weaknesses; some I believe need to be addressed before publishing this paper in a central stage such as EMNLP:

1. According to the authors, the HVI index is calculated based on the number of hallucinations of all the generated texts. However, it doesn’t take into consideration the length of the text. Technically speaking, an LLM that generates an empty string every time would get a high HVI score and that sounds a bit wrong. This may explain the locations of T0, T5 and XLNet on Figure 3. Also, the HVI index is calculated manually (using the annotated examples); calculating it for other LLMs is labor intensive and requires following the same annotation guidelines. This is a major limitation.

2. The high entropy replacement technique is quite simple and it may remove many named entities from the text, even if they are not hallucinated. Also, it is mentioned that the model considers consecutive high-entropy words together. But you didn’t mention what’s the criterion for being considered high entropy. I think that this important information should be provided in the paper and not in the appendix. Additionally, words are not tokens, how is that handled? this should be all described in the paper.

3. Since one of the main contributions of the paper is the dataset, I believe that some explanations on the annotation methodology and agreement levels should be provided in the paper and not only in the appendix.

4. The entailment exercise lacks some basic details. Only based on the information provided in the paper, it is hard to assess the efficacy of this technique. It should be removed from the paper, or provided with more empirical and analyzed information.

5. To me, there is a bit of overpromising in the first part of the paper, when discussing hallucination mitigation techniques. The techniques proposed in the paper are very simple and probably not very useful. The results shown for the high entropy technique, were calculated only for one model (GPT 3) and it is not clear what reduction of 10 HVI points actually mean in terms of how many hallucinations are removed. Also, it is not clear (and not discussed) whether the high entropy technique sometimes creates new hallucinations.


**Reproducibility:**

4: Could mostly reproduce the results, but there may be some variation because of sample variance or minor variations in their interpretation of the protocol or method.

**Reviewer Confidence:**

5: Positive that my evaluation is correct. I read the paper very carefully and I am very familiar with related work.

---

> ### Author Rebuttal · Authors · 2023-08-28
>
> Review 2:
>
> Kindly consult the general feedback regarding the "long abstract and long appendix," as well as the critique on mitigation techniques in the rebuttal for Reviewer 1.
>
> It doesn’t take into consideration the length of the text
> Thanks for pointing this out. We concur that the significance of the prompt's length is not the sole aspect to consider. Additionally, factors such as the intricacies related to formality and readability, the count of named entities (NEs), the presence of out-of-vocabulary words (OOVs) within the provided prompt, and the overall entropy or perplexity should all be taken into account when distinguishing between instances where the LLM is hallucinating vs. not hallucinating. We have indeed acknowledged the context length and hallucination tendency in Appendix C.1, page 18. The available space doesn't allow us to do justice to this issue which will take a longer treatment. A recent arxived paper [4] gives insights into what part of the prompt context is important - such as the start and the end, and empirically shown that most of the middle context of the prompt gets lost during the generation.
> LLM generates empty string
> Yes, although this is theoretically possible, we have never experienced empty string generation in our numerous experiments.
> HVI is calculated manually
> Certainly, we concur. However, we maintain that this aspect does not diminish the significance of this paper, which introduces the inaugural comprehensive metric for quantifying the hallucination vulnerability of a given LLM. Unfortunately, there is no existing measure as of today to quantify the hallucination vulnerability of a given LLM, and defining one such metric has great benefits towards policy making and in many other aspects such as having a uniform evaluation metric for any downstream NLP applications. Please refer to the introduction section of our paper where we discuss the details of the larger necessity of having a met to calculate hallucination vulnerability. In subsequent endeavors, we aim to broaden our research by developing a framework for the automated assessment of the hallucination vulnerability of a given LLM. However, delving into that aspect goes beyond the focus of this initial paper on the subject, which is our current contribution.
>
> High entropy replacement technique is simple and error prone
> Yes, we agree that the entropy-based method aka the black-box method has limitations. This method could be described as a precision model where the model can probably alleviate high hallucination-prone points. At the same time, it can replace named entities, low-frequency words, OOVs, etc. To handle this, we came up with a simple solution. We calculated the concreteness score for each word utilizing the previous research [5]. For example, in the following sentence from the available alternatives, we choose the word/token having the least concreteness score. In that way, we make sure that the content is getting generated is less risky.
>
> “The official stance of the [MASK] {MASK word was *USA*} on the Russia-Ukraine war has been consistent in supporting Ukraine’s sovereignty, territorial integrity, and the peaceful resolution of the conflict.”
> Options:
> US
> Government - we choose this as this word has least concreteness score
> America
> Country
>
> We just found another paper [6] (arxived after our submission to EMNLP, in August 2023) that follows a similar philosophy of investigating methods to de-concretize/de-formalize an LLM’s output. We will add the necessary description and discussion in the final version.
>
> Conversely, our gray-box approach leans towards recall orientation, while human-in-the-loop intervention results in an overall editing rate of 26%. Balancing precision and recall is currently slated as a follow-up endeavor.
>
> Words are not tokens
>
> —------------------------------
>
> We understand that some readers might misunderstand a token as a sub-word unit. We will rectify such wording(s). Please clarify if you are referring to this issue or something else we are unable to understand - please provide more details in that case.
>
> Details on entailment-based method
>
> —----------------------------------------------
>
> We have provided a detailed description of the experiment in Appendix G.2, page #26. However, we realized that we missed mentioning the threshold values based on what we have categorized as “refute”, “support”, and “not enough information”. Those values were experimentally chosen and we will add all the necessary details in the appendix in the final version of the paper. We will also add necessary parts in the main paper with the extra granted page post-acceptance.
>
> Overpromising in the first part of the paper
>
> —------------------------------------------------------
>
> We have mentioned in our contribution highlights on page #3 that “While complete mitigation can be a herculean task, we suggest two mitigation strategies to alleviate hallucination.” This should clarify that any future researchers interested in using the HILT dataset, HVI, and other resources produced should be able to compare their results with the two baselines provided in this paper. Indeed, there is a lot of room for improvement in both methods, but we believe that we're offering solid baselines that future researchers can compare with. Hallucination is a larger problem to be solved by the community, and hence it is beyond the scope of any one paper to cover all facets.
>
> Black-box entropy-based technique, were calculated only for one model (GPT-3)
>
> —-----------------------------------------------------------------------------------
>
> This is NOT accurate. We have conducted experiments for all the 15 LLMs. We have only reported GPT-3 in the main paper as we found that it is the LLM that has the highest HVI score. Experimental details and results for the other 14 LLMs are reported in Tables 9-23 in Appendix section G.1, page #22-25. We will be sure to draw attention to this in the main paper.
>
> It is not clear what reduction of 10 HVI points actually mean
>
> —----------------------------------------------------------------------------------
>
> We have provided further details in Table 24, page #27. We agree that readers might be interested in the actual numbers. So, we will add actual numbers/counts in Table 24 in addition to the HVI scores (this should be possible since we will have one extra page for the main paper post-acceptance). For example, for GPT-3, in the case of Silver Lining, for the “Numeric Nuisance” category, we have had a total of 850 instances in the gold test data consisting of 2000 sentences. After applying the entropy-based method there was no removal, but when we applied the gray-box method we saw a possibility of removing 82 cases. We will add these details in Table 24 for curious readers.
>
> Whether the high entropy technique sometimes creates new hallucinations.
>
> —----------------------------------------------------------------------------------------------------------
>
> That's an interesting question, and we were also skeptical about this possibility. To address this, we have investigated employing the concreteness score, which we did not include in the submission as experiments were ongoing at the time. Theoretically, replacing high entropy words can lead to new hallucinations, but we have not seen any new hallucination generated for any of the 2000 sentences in the test gold set. It is quite possible that it is there somewhere in the larger dataset. We will add relevant discussion to encourage future researchers to utilize the dataset and explore such possibilities. Moreover, we are interested in pursuing this direction as well. This leads to a question: What is the best possible replacement strategy? There are many open possibilities. Additionally, although our proposed mitigation techniques reduce hallucinations in some cases, we are not claiming that these methods remove hallucinations completely.
>
> URL in a footnote
>
> —-----------------------
>
> Thanks, we will make those changes in the final version.
>
> Taxonomy of hallucination types in the abstract
>
> —--------------------------------------------------------------
>
> Point is well taken, we will rewrite the abstract. Given the number of issues explored, we got carried away but we can focus on the main contributions in the abstract.
>
> Figure 2 was not mentioned in the text
>
> —--------------------------------------------------
>
> We will fix it and fix any errata of missing references.
>
> Discussion and Limitations
>
> —------------------------------------
>
> We have named our limitation discussion as “Discussion and Limitations”, although we can separate our discussion and limitation separately.
>
> FAQ section after the Reference section
>
> —-----------------------------------------------------
>
> FAQ provides a nice way to summarize necessary points to the readers. If you have any specific suggestion(s) for the FAQ section, we will be happy to incorporate them.
>
> REFERENCES:
> [4] Liu, Nelson F., Kevin Lin, John Hewitt, Ashwin Paranjape, Michele Bevilacqua, Fabio Petroni, and Percy Liang. "Lost in the middle: How language models use long contexts." arXiv preprint arXiv:2307.03172 (2023).
> [5] Brysbaert, Marc, Amy Beth Warriner, and Victor Kuperman. "Concreteness ratings for 40 thousand generally known English word lemmas." Behavior research methods 46 (2014): 904-911.
> [6] Varshney, Neeraj, Wenlin Yao, Hongming Zhang, Jianshu Chen, and Dong Yu. "A stitch in time saves nine: Detecting and mitigating hallucinations of llms by validating low-confidence generation." arXiv preprint arXiv:2307.03987 (2023).
>
> We hope the detailed response will allow you to consider if this paper deserves a higher score that is consistent with the other two reviewers. Hallucination is an important and complex problem that cannot be addressed in a single paper, but we emphasize that this paper brought in significant, timely, and early contributions in this intricate-but-important area.

---

### Official Review · Reviewer_3m5D · 2023-08-05

**Soundness:** 4

**Excitement:**

3: Ambivalent: It has merits (e.g., it reports state-of-the-art results, the idea is nice), but there are key weaknesses (e.g., it describes incremental work), and it can significantly benefit from another round of revision. However, I won't object to accepting it if my co-reviewers champion it.

**Missing References:**

This work does not acknowledge existing literature in hallucination. Some missing references:
[1] “Understanding Factuality in Abstractive Summarization with FRANK: A Benchmark for Factuality Metrics” - https://arxiv.org/abs/2104.13346 —> provides detailed categorisation of hallucination.
[2] Factuality Enhanced Language Models for Open-Ended Text Generation - https://arxiv.org/abs/2206.04624  —> first paper to use the concept of “factually correct prompt” vs “factually incorrect prompt”.


**Paper Topic And Main Contributions:**

This paper designs detailed categorization of hallucination in LLMs, and evaluates 15 LLMs based on this categorization. This paper also proposes a new metric that can be used to evaluate and rank LLMs based on their vulnerability to generate hallucinations.

**Questions For The Authors:**

1. How much was each annotator paid?
2. In Table 2, how was this “hallucination” level measured? It is unclear how this black-box approach is evaluated.


**Reasons To Accept:**

1. Fine-grain categorization of hallucination and evaluation of 15 LLMs based on this proposed categorization
3. Proposed a metric that measures the LLM’s hallucination level that allows for comparison between different LLMs. This paper ranks 15 LLMs based on their proposed metric.

**Reasons To Reject:**

1. Proposed categorization appears to be developed in isolation from other existing related works.
2. This paper is hard to understand without referring to the appendix. It would be beneficial for readers if introduction/abstract contents are shortened and have more important details in the later sections (e.g., Section 5)
3. Mitigation methods are not novel

**Reproducibility:**

2: Would be hard pressed to reproduce the results. The contribution depends on data that are simply not available outside the author's institution or consortium; not enough details are provided.

**Reviewer Confidence:**

4: Quite sure. I tried to check the important points carefully. It's unlikely, though conceivable, that I missed something that should affect my ratings.

---

> ### Author Rebuttal · Authors · 2023-08-28
>
> Response to all the reviewers
> -------------------------------------------
> Long abstract and long appendix
> -------------------------------------------
> This paper consists of several sections dealing with distinct and necessary topics/components, so we decided to write a longer appendix. Knowingly we wrote a longer abstract for review purposes only. Certainly, by focusing on the central topics and ignoring definitions and descriptions of hallucination classes in the abstract, we will reduce it in our final version.
>
> Mitigation methods
> -------------------------
> Although we have clearly pointed out our main contributions on page #3, we would like to emphasize here that our contributions are - i) the definition of hallucination taxonomy, ii) defining the HVI metric, iii) presenting experiments for 15 LLMs, iv) introducing HILT dataset, and finally, v) providing *baseline* mitigation techniques. For mitigation, we have presented two classes of techniques: a) black-box, aka the entropy-based method having better precision, and b) gray-box, aka retrieval-based method having better recall. We expect us and other researchers to utilize the HILT dataset, HVI, etc., and compare new results with these baseline mitigation techniques. There is no doubt that these mitigation techniques are *NOT FOOLPROOF* and there are many other ways they could be improved. However, we explicitly mentioned on page 3 regarding our coverage: “While complete mitigation can be a herculean task, we suggest two mitigation strategies to alleviate hallucination.” [This sentence will be modified with an indication of the strategies for future enhancement. One paper is not sufficient to cover hallucination taxonomy, a manually annotated dataset, a metric to measure hallucination, experiments on 15 LLMs (our main contributions), and proposing comprehensive techniques for hallucination mitigation.]
>
>
> Review 1:
>
> Proposed categorization appears to be developed in isolation from other existing related works.
> We did not fully get this question/remark. We have acknowledged that previous researchers defined intrinsic and extrinsic kinds of hallucination and we have extended our definition drawing inspiration from those cited works. Thanks for highlighting [1]. This paper has also discussed about “factually correct” prompt” vs. “factually incorrect prompt”. We've assigned labels like "factual mirage" and "silver lining" to categorize such instances. Finally, we have defined six fine-grained classes to annotate hallucination at the sentence level. The work by [2] discussed hallucination in terms of name-nationality hallucination which is a subproblem of our definition of the “Generated Golem” hallucination category. So, we contend that our definitions are more inclusive and encompass other components observed in the literature. We would gladly incorporate any additional research outlining different categories of hallucinations that have been published subsequent to our submission. If you believe such research merits inclusion and discussion in this paper, please let us know.
>
> We will provide cost analysis of our annotation process.
> We have provided significant details on the annotation process in Appendix B. Regarding cost, we have experimented with several numbers. An honorarium of 10 INR (given majority of AMT annotators are from India) was offered per sentence due to the inherent complexity, which was ensured to be appropriate considering the 160-750 INR minimum wage/day (based on the Minimum Wages Act, 1948 [3]) in India (where the annotations were done) based on the average number of annotations across all annotators per day. Typically, in order to get annotation from the AMT platform people start with a lower compensation and finalize the best annotators, and then allow the best annotators to pay a higher amount to get the work done. We followed a similar philosophy to finalize annotators for our hallucination annotation task. We will further add a description in the appendix and main paper as to how we came up with such a compensation number.
>
> How this black-box approach is evaluated.
> Output from the black box method was evaluated manually. We have provided details in Appendix G.3 - pages #26-27.
>
> Missing references -  Thank you for pointing them out, we will include them in the final revision.
>
> REFERENCES:
> [1] Lee, Nayeon, Wei Ping, Peng Xu, Mostofa Patwary, Pascale N. Fung, Mohammad Shoeybi, and Bryan Catanzaro. "Factuality enhanced language models for open-ended text generation." Advances in Neural Information Processing Systems 35 (2022): 34586-34599.
> [2] Ladhak, Faisal, Esin Durmus, Mirac Suzgun, Tianyi Zhang, Dan Jurafsky, Kathleen Mckeown, and Tatsunori B. Hashimoto. "When Do Pre-Training Biases Propagate to Downstream Tasks? A Case Study in Text Summarization." In Proceedings of the 17th Conference of the European Chapter of the Association for Computational Linguistics, pp. 3198-3211. 2023.
> [3]  List of countries by minimum wage, https://en.wikipedia.org/wiki/List_of_countries_by_minimum_wage

---

### Meta-Review · Area_Chair_e6HQ · 2023-09-16

**Recommendation:** 5

**Metareview:**

This paper examines the hallucination problem by categorizing these by their category (acronym ambiguity, numeric nuisance, generated golem, virtual voice, geographic erratum, time wrap), degree (mild, moderate, alarming), and orientation (factual mirage or silver lining). They then provide two methods to mitigate these issues. The authors provide a sample of 75k examples of these hallucinations with 2k annotated samples. They also come up with an index (HVI) that allows them to rank LLMs by their likelihood of generating hallucinations. The reviewers agree that the paper is sound and are generally excited about the work. They provide several valid criticisms regarding the limitations of the HVI metric and somewhat simple mitigation strategies. However, this paper has a wide array of contributions that should be beneficial for the community. The fine-grained types of hallucination and annotations will probably be the most valuable component, though future work may build off of the other contributions. It is reasonable that the scope of a paper cannot include everything that the reviewers have brought up, though I think their suggestions will be a helpful guide for future work.

---

### Decision · Program_Chairs · 2023-10-07

**Decision:**

Accept-Main

**Comment:**

This paper examines the hallucination problem by categorizing these by their category (acronym ambiguity, numeric nuisance, generated golem, virtual voice, geographic erratum, time wrap), degree (mild, moderate, alarming), and orientation (factual mirage or silver lining). They then provide two methods to mitigate these issues. The authors provide a sample of 75k examples of these hallucinations with 2k annotated samples. They also come up with an index (HVI) that allows them to rank LLMs by their likelihood of generating hallucinations. The reviewers agree that the paper is sound and are generally excited about the work. They provide several valid criticisms regarding the limitations of the HVI metric and somewhat simple mitigation strategies. However, this paper has a wide array of contributions that should be beneficial for the community. The fine-grained types of hallucination and annotations will probably be the most valuable component, though future work may build off of the other contributions. It is reasonable that the scope of a paper cannot include everything that the reviewers have brought up, though I think their suggestions will be a helpful guide for future work.